# Glutathione Plays a Positive Role in the Proliferation of *Pinus koraiensis* Embryogenic Cells

**DOI:** 10.3390/ijms232314679

**Published:** 2022-11-24

**Authors:** Fang Gao, Chunxue Peng, Yue Zhang, Hao Wang, Hailong Shen, Ling Yang

**Affiliations:** 1State Key Laboratory of Tree Genetics and Breeding, School of Forestry, Northeast Forestry University, Harbin 150040, China; 2Institute of Biotechnology, Jilin Provincial Academy of Forestry Sciences, Changchun 130033, China; 3Engineering Technology Research Center of Korean Pine, State Forestry and Grassland Administration, Harbin 150040, China

**Keywords:** embryogenic callus proliferation, Korean pine, metabolomics, redox metabolism, somatic embryogenesis, transcriptomics

## Abstract

In the large-scale breeding of conifers, cultivating embryogenic cells with good proliferative capacity is crucial in the process of somatic embryogenesis. In the same cultural environment, the proliferative capacity of different cell lines is significantly different. To reveal the regulatory mechanism of proliferation in woody plant cell lines with different proliferative potential, we used Korean pine cell lines with high proliferative potential 001#–001 (Fast) and low proliferative potential 001#–010 (Slow) for analysis. A total of 17 glutathione-related differentially expressed genes was identified between F and S cell lines. A total of 893 metabolites was obtained from the two cell lines in the metabolomic studies. A total of nine metabolites related to glutathione was significantly upregulated in the F cell line compared with the S cell line. The combined analyses revealed that intracellular glutathione might be the key positive regulator mediating the difference in proliferative capacity between F and S cell lines. The qRT-PCR assay validated 11 differentially expressed genes related to glutathione metabolism. Exogenous glutathione and its synthase inhibitor L-buthionine-sulfoximine treatment assay demonstrated the positive role of glutathione in the proliferation of Korean pine embryogenic cells.

## 1. Introduction

Due to the rapid growth of population and sustained growth of the economy, global demand for wood continues to grow [1]. High-quality seedlings could increase forest productivity. Therefore, it is necessary to breed large numbers of seedlings with genetic superiority [2]. Asexual propagation by somatic embryogenesis could obtain more high-quality plantles [3,4]. Especially for long reproductive cycle pine species, somatic embryogenesis could contribute to rapid production of a large number of regenerated plants with stable characteristics, which has great applicative value [5]. Somatic embryogenesis is an effective model system to study morphological, physiological, molecular, and biochemical aspects of embryonic development of trees [6,7,8]. In the past decades, research on conifer somatic embryogenesis was mainly focused on the regulation technology and mechanism in the embryogenic callus (EC) induction stage and somatic embryo maturation stage [9,10,11]. In the large-scale breeding of conifers, the cultivation of EC with good proliferative ability is crucial in the whole process of somatic embryogenesis [12]. The proliferative potential of different conifer cell lines varies greatly, but the regulatory mechanism affecting the difference is still unclear. Therefore, an in-depth study on the regulatory mechanism that causes the difference in EC proliferative ability is of great significance to optimize and improve the technical system in conifer somatic embryogenesis.

Transcriptomics [13,14,15,16] and metabolomics [17,18] have been successfully applied to study developmental processes, including somatic embryogenesis [18,19]. Transcriptomics and metabolomics studies on somatic embryogenesis in conifer species mainly focused on somatic embryo maturation [19] and germination [18]. Park et al. [17] found that levels of metabolites, including some xanthosine and methyloxazole, were significantly different in an embryogenic and a non-embryogenic callus of hybrid loblolly pine (*Pinus rigida* × *P. taeda)*. Businge et al. [20] reported that sugar signaling affected the early induction of somatic embryos in *Picea abies*. In addition, tryptophan was only presented in two cell lines that can differentiate into somatic embryos. They further inferred that exogenous auxin was crucial to EC proliferation and somatic embryos differentiation. Further research on the regulatory mechanism of EC proliferation in conifers needs to be carried out urgently.

Korean pine (*Pinus koraiensis* Sieb. et Zucc.) is a dominant tree species of temperate zone climax community broad-leaved forest. It is also an extremely important high-quality precious timber tree and nut species and economic forest species in East Asia. The somatic embryogenesis technology system of Korean pine has been established previously [21,22,23,24]. The EC proliferation process is not only an important step in somatic embryogenesis but also an important material for large-scale production and genetic transformation [12]. Studies have found that the proliferative ability of different Korean pine cell lines is significantly different in the same cultural environment, but the regulatory mechanism that causes this difference is still unclear. We used two EC cell lines with high and low proliferative potential as tested materials and performed transcriptomic and metabolomic analyses. The results of transcriptome and metabolome assays were further confirmed by an exogenous addition and an intracellular content determination of glutathione. These results provide a scientific basis for improving the proliferative ability of embryogenic cells and improving the efficient plant regeneration system of Korean pine.

## 2. Results

### 2.1. Proliferative and Cytological Observation of Cell Lines with Different Proliferative Potential

The surface morphology of Fast (Fast proliferative potential cell line is abbreviated as F) and Slow (Slow proliferative potential cell line is abbreviated as S) cell lines was different. The F cell line had abundant filamentous early somatic embryos on the surface (Figure 1a) and a relatively fast proliferative rate (the proliferative rate was 740.0 ± 52.0 a), which EC was transparent as a whole. While the S cell line had poor filamentous early somatic embryos (Figure 1b), a few brown spots on the surface of the callus, and a relatively slower proliferative rate (the proliferative rate was 425.0 ± 20.2 b, which was 57.4% of the F cell line). A large number of early somatic embryos (Figure 1c,d) was seen by cytological observation. It was found that the edges of the embryo head (eh) and the embryo suspensor (es) of the F cell line were clearer than those of the S cell line.

### 2.2. Transcriptome Analysis

#### 2.2.1. RNA Sequencing and Transcriptome De Novo Assembly

The F and S cell lines (six samples in total) were sequenced by Illumina Hiseq platform. A total of 40 Gb clean data was obtained. The clean data in each sample were more than 6 Gb, and the base ratio of Q30 was 93.08–95.93%. The impurities of reads were removed to obtain clean reads. The GC content was more than 43.78%, which met the requirements of subsequent transcriptome de novo assembly. A total of 16,027 unigenes was obtained from de novo assembly, with high assembly integrity. The mapped ratio values of the six samples reached above 75.35%, so the result of de novo assembly in six samples was good and met the requirements for further tests.

#### 2.2.2. Screening of Differentially Expressed Genes and Functional Annotation

There were 1883 differentially expressed genes (DEGs) between F and S cell lines, in which 870 genes were upregulated and 1013 genes were downregulated (FDR < 0.01, FC ≥ 2). A total of 1422 DEGs with annotation information was obtained by BLAST (E-value ≤ 1 × 10^−5^) and HMMER (E-value ≤ 1 × 10^−10^). Principal component analysis (PCA) was performed on six samples to understand the correlation of overall DEGs between the two samples of F and S cell lines and the difference of DEGs between F and S cell lines. According to the PCA diagram, there was little difference in the principal components in the samples of F and S cell lines (Figure 2), while there were significant differences in the principal components between the samples of F and S cell lines, indicating that there were significant differences in DEGs between the samples of F and S cell lines. The DEGs sequences were compared with COG, GO, KEGG, KOG, Pfam, Swiss-prot, eggNOG, and Nr databases using DIAMOND software, and the annotation results of unigene in KEGG were obtained using KOBAS. The number of annotated unigenes in COG, GO, KEGG, KOG, Pfam, Swiss-prot, eggNOG, and Nr databases was 407, 1139, 842, 540, 1123, 924, 1400, and 1148, accounting for 21.61%, 60.49%, 44.72%, 28.68%, 59.64%, 40.07%, 74.35%, and 60.97% of the DEGs, respectively.

#### 2.2.3. Screening of Differentially Expressed Genes and Functional Annotation

A total of 1139 DEGs in this study was distributed in 46 GO items (Appendix A at International Journal of Molecular Sciences online). Upon GO classification based on biological processes, more DEGs have been annotated in “response to stimulus”, “metabolic process”, “single-organism process”, and “cellular process”. Among the molecular functions, the number of DEGs annotated by “binding”, “transporter activity”, “catalytic activity”, “nucleic acid binding transcription factor activity”, and “antioxidant activity” was abundant. In the categories of cellular components, the number of DEGs annotated by “membrane”, “cell”, “cell part”, and “membrane part” was abundant.

#### 2.2.4. COG Classification of Differentially Expressed Genes

The COG database was used to compare with DEGs, and the COG annotations were classified with a total of 22 functions (Appendix A at International Journal of Molecular Sciences online), and 492 genes were annotated, accounting for 26.13% of the DEGs. Secondary metabolites biosynthesis, transport and catabolism, general function prediction, carbohydrate transport and metabolism, signal transduction mechanism, posttranslational modification, protein turnover, chaperones, lipid transport and metabolism, cell wall/membrane/envelope biogenesis, signal transduction mechanisms, and other functions account for a high proportion.

#### 2.2.5. KEGG Enrichment Analysis in Differentially Expressed Genes

The DEGs were annotated in 115 pathways and were mainly enriched in plant-pathogen interaction, phenylpropanoid biosynthesis, MAPK signaling pathway-plant, plant hormone signal transduction, protein processing in the endoplasmic reticulum, carbon metabolism, glutathione (GSH) metabolism, ascorbate and aldarate metabolism, pyrimidine metabolism, and other pathways (Appendix A at International Journal of Molecular Sciences online).

##### KEGG Enrichment Analysis in Differentially Expressed Genes

A total of 17 GSH-related DEGs was significantly changed between F and S cell lines (Figure 3), including eight *GST* family genes, in which seven genes were downregulated and one gene was upregulated. Three *GULO* family genes were downregulated. The expression of *CARP*, *ODC1*, and *DHAR* family genes was downregulated. Additionally, there were two *AAO* family genes, in which one gene was downregulated, and the other was upregulated. One *PGD* family gene trended toward upregulation. In general, most of the DEGs related to GSH between F and S cell lines (14) trended toward downregulation, and only a few differential genes (3) trended toward upregulation.

##### Analysis of Hormone Metabolism-Related Genes

Through the observation of transcriptome data, it was found that many DEGs were involved in a plant hormone signal transduction pathway, mainly involved in the metabolic pathway of indole-3-acetic acid (IAA) and cytokine in biosynthesis. Therefore, in this study, genes that played a role in the anabolism, transport, and signal transduction of these two hormones were selected for specific analysis.

##### Specific Analysis of Auxin-Related Genes

There were seven auxin-related DEGs between F and S cell lines that were observed, including three genes of *SAUR* family (Figure 4), in which the expression of two genes trended toward upregulation and one gene trended toward downregulation. In addition, *GH3*, *AUX1*, *JAZ*, and *ARF* genes in the family were trended toward downregulation.

**Figure 4 ijms-23-14679-f004:**
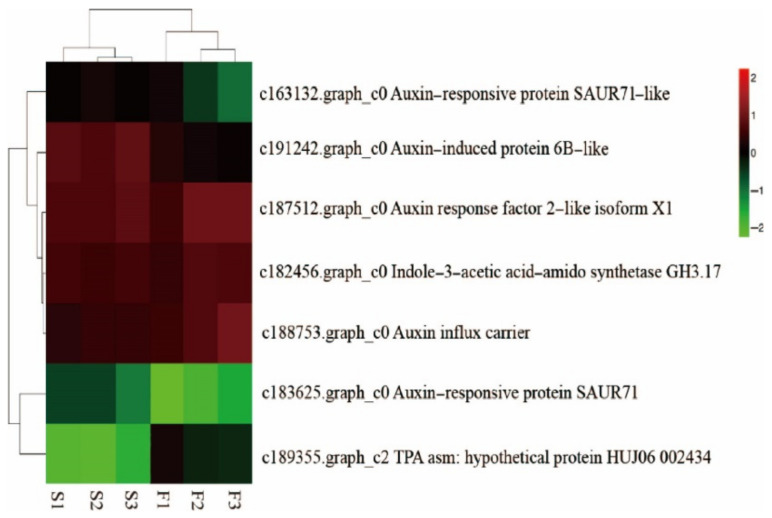
Cluster heat map of auxin—related differential genes between F and S cell lines of Korean pine.

##### Analysis of Cytokinin-Related Genes

Two differentially expressed genes related to cytokinin existed between F and S cell lines, the *AHK2_3_4* family genes were downregulated, and the *ARR*-*B* family gene was upregulated (Appendix A, supplementary data at International Journal of Molecular Sciences online).

### 2.3. qRT-PCR

Twelve DEGs related to GSH and auxin between F and S cell lines of Korean pine were selected for qRT-PCR verification (Appendix A, supplementary data at International Journal of Molecular Sciences online). The gene functions were shown in Figure 5. The results showed that three DEGs were upregulated, and nine DEGs were downregulated. Except for the *GST* family gene (c187045.graph_c0), the expression trends of the remaining 11 DEGs were similar to the RNA-Seq results, indicating that the differential genes obtained by sequencing in this study were credible.

**Figure 5 ijms-23-14679-f005:**
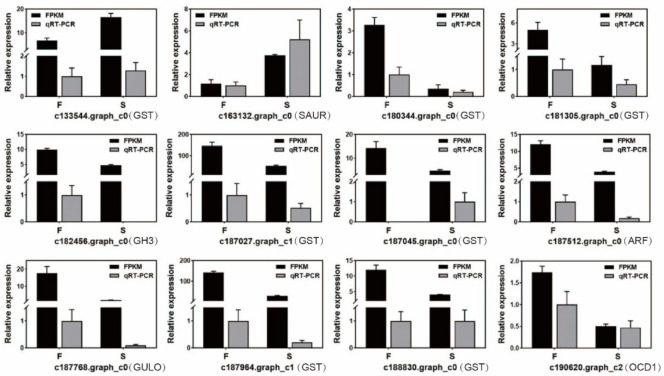
qRT-PCR analysis of differentially expressed genes between F and S cell lines of Korean pine. Note: The average three biological replicates were counted for each treatment, and ANOVA was performed on the data in the figure, mean ± se.

### 2.4. Metabolomics Analysis in Cell Lines of Korean Pine with Different Proliferative Potentials

A total of 893 metabolites was obtained from F and S cell lines (six samples in total). Principal component analysis (PCA) was performed on six samples to understand the correlation of overall metabolisms between the two samples of F and S cell lines and the difference of metabolites between F and S cell lines. According to the PCA diagram, there was little difference in the principal components in the samples of F and S cell lines (Figure 6), while there were significant differences in the principal components between the samples of F and S cell lines, indicating that there were significant differences in metabolites between the samples of F and S cell lines.

**Figure 6 ijms-23-14679-f006:**
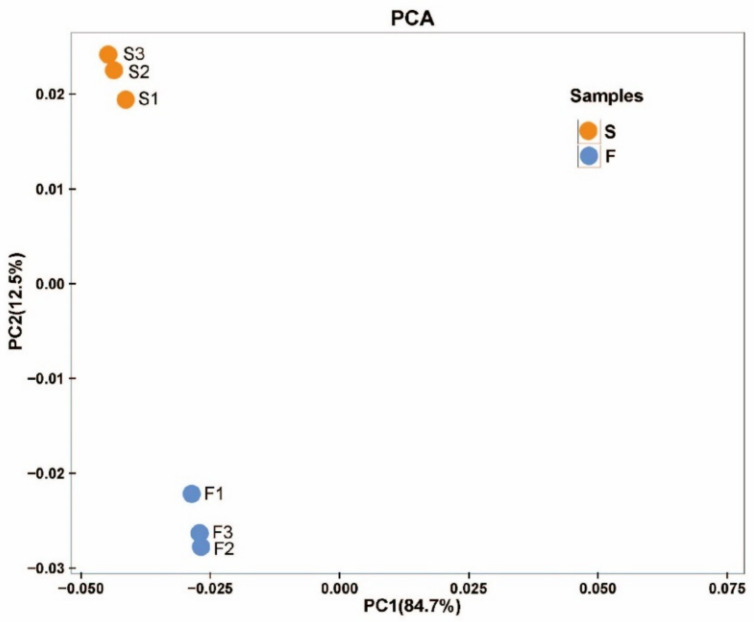
PCA analysis of the principal components in F and S cell lines of Korean pine. Note: The X-axis represents the first principal component, and the Y-axis represents the second principal component.

#### 2.4.1. Differential Metabolite Screening

Volcano plot revealed 412 differential metabolites (DEMs) between F and S cell lines, in which 183 metabolites were upregulated and 229 were downregulated (FC > 1, *p* < 0.05, and VIP > 1). All DEMs were divided into 27 categories, mainly 31 kinds of carboxylic acids and their derivatives; 17 kinds of fatty acyl compounds; 11 kinds of organic oxygen compounds; 6 kinds of imidazopyrimidines; 6 kinds of purine nucleosides and organic compounds; 4 kinds of nitrogen compounds; as well as 3 kinds of benzene and its substitutes, indole and its derivatives, phenols; purine nucleotides; pyrimidine and its derivatives, pyrimidine nucleosides, and pyrimidine nucleotide metabolites; 2 kinds of cinnamaldehyde, cinnamic acid and its derivatives and flavonoid metabolites; and a kind of (5’->5’)-dinucleotide, 5’-deoxyribonucleoside, biotin, and its derivatives.

#### 2.4.2. KEGG Functional Annotation and Enrichment Analysis of DEMs

KEGG pathway map showed that DEMs were enriched in ABC transport, 2-oxocarboxylic acid, amino acid biosynthesis, aminoacyl tRNA biosynthesis, linoleic acid metabolism, and purine metabolism pathways accounted for the top six (Figure 7). In addition, glucosinolate biosynthesis, GSH metabolism, ascorbate and alginate metabolism, pyrimidine metabolism, and phytohormone signaling pathways also accounted for a high proportion. There were no DEMs related to auxin and cytokinin between F and S cell lines. GSH metabolic pathway accounted for a high proportion in the KEGG pathway, and there were nine differentially expressed metabolites (Figure 8), in which L-ornithine, L-cysteine, spermidine, L-glutamate, oxidized glutathione (GSSG), N-acetyl-D-glucosamine, spermine, inositol, and S-methylglutathione were trended toward downregulation.

**Figure 7 ijms-23-14679-f007:**
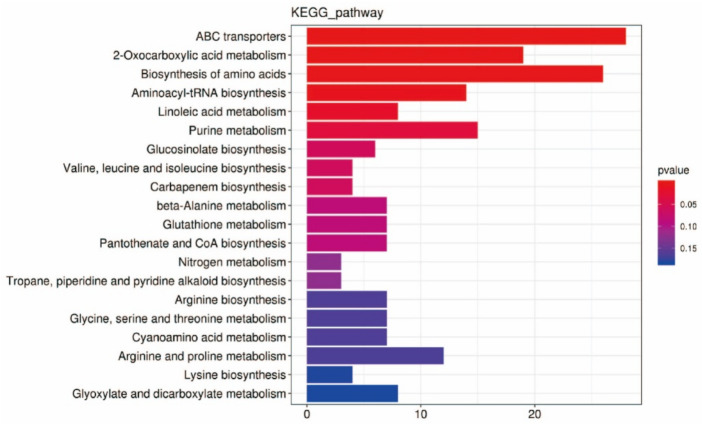
Classification of the KEGG pathway in DEMs in F and S cell lines of Korean pine.

**Figure 8 ijms-23-14679-f008:**
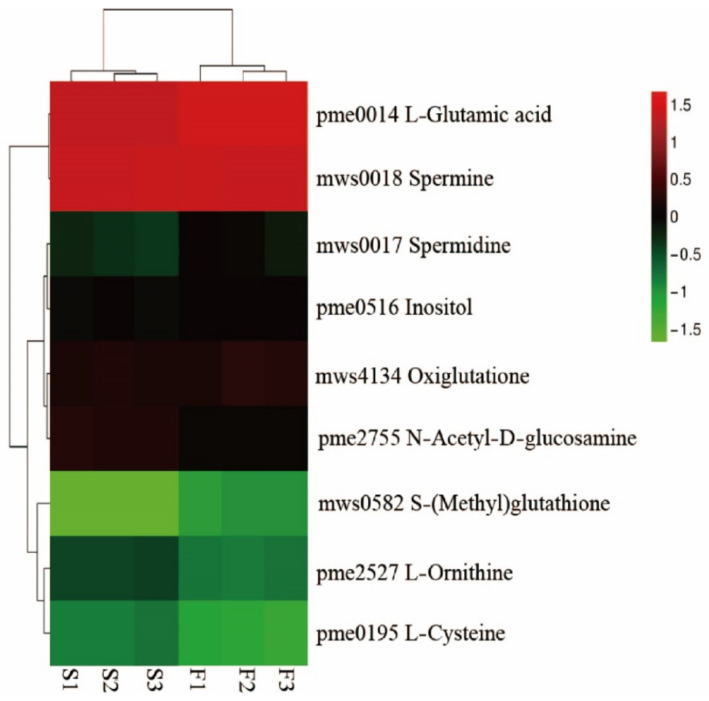
Cluster heat map of GSH-related DEMs between F and S cell lines of Korean pine.

#### 2.4.3. Combined Analysis of Metabolomics and Transcriptomics

The selected genes and metabolic pathways (*p* < 0.05) were preferentially analyzed. The KEGG co-enrichment results are shown in Figure 9. It could be seen that the differential expression genes and metabolites were co-enriched in 68 KEGG pathways. Among them, phenylpropane biosynthesis, α-linolenic acid metabolism, glycerophospholipid metabolism, GSH metabolism, glyoxylate and dicarboxylic acid metabolism, plant hormone signal transduction, cysteine and methionine metabolism, carbon metabolism, glycolysis/gluconeogenesis, and amino sugar and nucleotide sugar metabolism pathways accounted for a high proportion.

#### 2.4.4. Canonical Correlation Analysis in GSH-Related Genes and Metabolites

Canonical correlation analysis (CCA) was performed on the differential genes and metabolites related to GSH for further exploring the gene most related to metabolites (Figure 10). It was found that a total of 17 differential genes were related to GSH, including 8 genes in *GST* family, 3 genes in *GULO* family, 2 genes in *E1.10.3.3* family, and 1 gene in each of *CARP* families, *ODC1*, *DHAR* and *PGD*. Nine DEMs were related to GSH, namely L-ornithine, L-cysteine, spermidine, L-glutamic acid, oxidized GSH, N-acetyl-D-glucosamine, spermine, S-methylglutathione, and inositol. CCA shows that the *PGD* family gene and the *GST* family gene were not in the same region and were far away from other differential genes and metabolites, indicating that these two genes had a low correlation with other genes and metabolites, while other genes and metabolites had high correlation.

### 2.5. Verification

#### 2.5.1. Determination of Intracellular GSH in F and S Cell Lines

The intracellular GSH content of F and S cell lines was shown in Figure 11a–d, which was consistent with the analysis and speculation of transcriptome and metabolome data. The intracellular GSH (Figure 11a) and total glutathione (T-GSH) (Figure 11b) contents of the F cell line were significantly higher than those of the S cell line. The content of GSSG (Figure 11c) and the ratio of GSH/T-GSH (Figure 11d) in the F cell line were higher than those in the S cell line, but the difference was not significant. The present results validated the results of transcriptomic and metabolomic analyses.

#### 2.5.2. Determination of Intracellular Hormone in F and S Cell Lines

The results of intracellular hormone content measurement in F and S cell lines are shown in Figure 11e–j. The intracellular IAA content in the F cell line was higher than that in the S cell line (Figure 11e), but the difference was not significant (*p* > 0.05). In addition, the intracellular zeatin (ZR) (Figure 11f) and abscisic acid (ABA) (Figure 11g) contents in F and S cell lines were also not significant. The effects of different phytohormones on plant physiological activities were both mutually constraining and promoting. IAA and ZR were growth-promoting factors during plant growth and development, while ABA inhibits cell growth and promotes maturation. Therefore, the ratios of IAA and ZR to ABA were used to evaluate the intracellular hormone ratios of F and S cell lines, and it was found that IAA/ABA (Figure 11h), ZR/ABA (Figure 11i), and (IAA + ZR)/ABA (Figure 11j) were significantly higher in the F cell line than in the S cell line, which suggested that the ratio of growth-promoting hormones was higher in the F cell line than in the S cell line, indicating that various intracellular hormones acted synergistically to control the growth of Korean pine EC.

#### 2.5.3. Validation of the Effect of Exogenous GSH and L-Buthionine-Sulfoximine (BSO) on the EC Proliferation Rate of F and S Cell Lines

Exogenous addition of GSH promoted EC proliferation in F and S cell lines, and exogenous BSO inhibited EC proliferation (Figure 11k,l), which further validated the results of transcriptomics and metabolomics analyses.

## 3. Discussion

The proliferative process of Korean pine EC was the result of the dynamic balance of embryonic cell division and differentiation. This process was closely related to GSH and plant growth regulators, which was consistent with Belmonte et al. [25]. In the following discussion, the analysis of differential genes and metabolites related to GSH, auxin, and cytokinin was mainly discussed.

### 3.1. Regulation of GSH Related Genes and Their Metabolites on Cell Proliferation

GSH is an important antioxidant in plants, which can protect cells from oxidative stress [26]. It plays a significant role in biosynthetic pathways, antioxidant biochemistry, and redox homeostasis. The transformation of the GSH redox state affected many functions of cells, including the number and morphological characteristics of somatic embryos [27,28]. GSH regulates gene expression and cell division and differentiation in different organisms [29]. It was found that there were 17 differential genes related to GSH between F and S cell lines. The GSH cycle pathway in the two cell lines of Korean pine is shown in Figure 12.

Genes of the *GST* family regulated GSH S-transferases. *GST* is a family of proteins with various forms and similar three-dimensional structures. GSH S-transferase is a key enzyme in the GSH binding reaction and possesses a well-defined GSH binding domain at its active site, which catalyzed the initial step of GSH binding reaction [30]. The combination of GSH and GST (RX (Organic halide) + GSH - (GST) → HX (Halo acid) + GS − R (R − S-glutathione)) improved the stress resistance of plants and detoxified them. There were eight differentially expressed genes in the *GST* family, most of which were downregulated, and only one gene was upregulated. Therefore, it was speculated that the stress resistance of the F cell line was higher than that of the S cell line.

GSH dehydrogenase/transferase family genes are regulated by dehydroascorbate reductase. Dehydroascorbate reductase (DHAR) is an important antioxidant enzyme in plants. It is the key enzyme to promote ascorbic acid (ASA) regeneration in the process of ASA-GSH cycle, and it plays an important role in protecting cell components against oxidative damage [30]. DHAR catalyzes GSH and dehydroascorbate (DHA) to produce GSSG and ascorbate (ASA) and regulates the ratio of ASA and DHA. DHAR uses ASA to maintain the normal metabolic level of ASA in plants and plays a significant role in resisting cellular oxidative damage. The activity of DHAR increased with the increase in intracellular GSH content [31]. There was one differentially expressed gene related to the *DHAR* family. Its expression in the F cell line was higher than that in the S cell line, and its expression was downregulated in the S cell line. This further suggested that the stress resistance of the F cell line might be higher than that of the S cell line, which needed to be further investigated.

*CARP* is a leucine peptidase family gene. The overexpression of CARP increased the catalytic activity of several aminopeptidase substrates [32]. CARP acts on small peptides produced by protein degradation or provided by exogenous sources to produce amino acids for further cell metabolism [33]. In this study, compared with the F cell line, the expression of *CARP* family gene in the S cell line trended toward downregulation. In the GSH metabolic pathway, γ-glutamylcysteine, glycine (Gly), and ATP were catalyzed by GSH synthase (GSS) to form GSH, ADP, and P1 [34]. Therefore, it was speculated that the GSH synthesis pathway of Korean pine F cell line was conducive to the accumulation of GSH, followed by the S cell line. Further metabonomic analysis showed that the results were consistent with those of transcriptomic analysis, which revealed that intracellular GSH was an important participant in the difference of proliferative potential between different embryogenic cell lines of Korean pine. The determination of intracellular GSH content in F and S cell lines of Korean pine confirmed this hypothesis. It was found that exogenous GSH promoted the proliferation and BSO inhibited the proliferation of Korean pine EC cells. The results of transcriptome and metabolome assays were further confirmed by an exogenous addition and an intracellular content determination. These results provide a scientific basis for improving the proliferative ability of embryogenic cells and improving the efficient plant regeneration system of Korean pine, revealing the regulatory mechanism of EC proliferation in conifers and laying a foundation for large-scale breeding of Korean pine.

### 3.2. Regulation of Auxin-Related Genes and Their Metabolites on Cell Proliferation

An auxin signal transduction pathway between F and S cell lines in Korean pine is shown in Figure 13. Indole-3-acetic acid (IAA) is the main form of auxin in higher plants, which is synthesized by an indole precursor of the tryptophan amino acid biosynthesis pathway. In the past few decades, it has been clearly proved that auxin had important effects on plants at the molecular level [35]. *AUX1* family genes are auxin influx carriers that mediate the transport of amino-acid signaling molecules. *AUX1* family gene restored auxin reactivity and promoted growth [36]. Auxin signaling was sensed by the *TIR1* family proteins, and binding of auxin to TIR1 can stabilize the interaction between *TIR1* and Aux/IAA, thus leading to ubiquitination of Aux/IAA and binding to DNA for regulating auxin-related gene expression [37]. *GH3* is an early gene regulated by auxin [38]. *GH3* family genes catalyzed free IAA into IAA amino acids and regulated the level of endogenous auxin by regulating the conversion of free and conjugated auxin.

The plant hormone auxin modulates cell proliferation and cell expansion in part by changing gene expression. Primary auxin response genes consist of members of three gene families: Aux/IAA, GH3, and SAUR [39]. The IAA regulates many aspects of plant growth and development. These processes are controlled by auxin-mediated changes in cell division, expansion, and differentiation [40]. This study found that the expression of the Aux/IAA gene trended toward downregulation in the S cell line compared with the F cell line, and the Aux/IAA gene promoted cell expansion [40]; therefore, this may be one of the reasons for the rapid proliferation of F cell lines. Auxin response factor (*ARF*) family genes are auxin response factors. A high concentration of endogenous IAA is conducive to binding with receptors and releasing *ARF* [41]. The expression of the *ARF* gene in the S cell line trended toward downregulation, indicating that the receptor role of *TIR1* in auxin signal transduction might be weakened, and the gene expression mode of the F cell line was conducive to the accumulation, transportation, and signal transmission of IAA level. Therefore, it was considered that the level of IAA in the F cell line might be higher than that in the S cell line, and IAA was involved in the regulation of EC proliferative ability in Korean pine. However, there was no significant difference in the expression of auxin-related metabolites between F and S cell lines. By measuring the intracellular hormone content, there was no significant difference in the contents of IAA, ZR, and ABA in F and S cell lines, but IAA/ABA, ZR/ABA and (IAA + ZR)/ABA were significant differences. Therefore, it can be seen that the proportion of growth-promoting hormones in the F cell line was higher than that in the S cell line. The proliferation of Korean pine EC was controlled by the synergistic effect of various hormones in cells.

## 4. Materials and Methods

### 4.1. Collection and Disinfection of Plant Materials

Full sibling family cones 1# were collected from a Korean pine seed orchard of Lushuihe Forestry Bureau of Heilongjiang Province 1# on 1 July 2018. The cones were washed with 50-fold dilution detergent for 30 min, rinsed continuously with running water for 8 h, and then placed in an ultra-clean workbench for sterilization with 75% alcohol for 50 min. Then, they were taken out, soaked in 75% alcohol for 1 min, and washed 3–5 times with sterile distilled water.

### 4.2. Embryogenic Callus Induction and Proliferation

Korean pine megametophyte was used as an explant and put in EC induction medium. The EC induction medium was mLV [42] basic medium supplemented with L-glutamine (500 mg·L^−1^), sucrose (30 g·L^−1^), acid hydrolyzed casein (500 mg·L^−1^), 1-Naphthylacetic acid (2 mg·L^−1^), 6-benzylaminopurine (1.5 mg·L^−1^), and gelrite (4 g·L^−1^) (gelrite, Sigma Aldrich, St Louis, MO, USA). Five explants were inoculated into each cultural dish (90 mm diameter, 20 mm depth). After adjusting pH to 5.8, we carried out high-temperature and high-pressure sterilization (121 °C, 20 min), cooled the cultural medium to 55–60 °C, and added L-glutamine by filtration sterilization (0.22 μm filter). After inoculating, the cultural dishes with explant were cultured in the dark at 23 ± 2 °C.

When the EC cluster grew to a diameter of 1–2 cm, it was transferred to the proliferation medium. The medium was mLV basic medium supplemented with sucrose (30 g·L^−1^), 2,4-dichlorophenoxyacetic acid (2.26 μmol·L^−1^), 6-benzylaminopurine (0.44 μmol·L^−1^), acid hydrolyzed casein (500 mg·L^−1^), L-glutamine (500 mg·L^−1^), and gelrite (4 g·L^−1^). The EC was sub-cultured every two weeks. The initial inoculation amount of EC was 0.2 g, 3 replicates per cell line, and the cultural conditions were the same as those of induction culture.

### 4.3. Proliferative Rate Calculation and Cell Observation of EC

On the 14th day of the proliferative culture, the weight was measured, the proliferative rate was calculated, and the cytological observation was carried out.
$$\mathrm{EC}\;\mathrm{proliferative}\;\mathrm{rate}\;(\%)=\frac{\mathrm{Fresh}\;\mathrm{weight}\;\mathrm{of}\;\mathrm{EC}\;\mathrm{after}\;\mathrm{proliferation}\;-\;\mathrm{Fresh}\;\mathrm{weight}\;\mathrm{of}\;\mathrm{inoculated}\;\mathrm{EC}}{\mathrm{Fresh}\;\mathrm{weight}\;\mathrm{of}\;\mathrm{inoculated}\;\mathrm{EC}}\;\times \;100$$

The cell lines, 001#–001 (F, Fast proliferative potential cell line is abbreviated as F) with high proliferative potential and 001#–010 (S, Slow proliferative potential cell line is abbreviated as S), with low proliferative potential. The cell lines F and S were used for cytological observation. The method used was as follows: fresh EC was put on the slide, stained with 0.1% safranine dye for 10 min, and covered with a cover glass. The cover glass was gently tapped to spread the plant tissues evenly. Then, it was immediately observed and recorded under an optical microscope (Olympus CX 31, Tokyo, Japan; equipped with Canon ds126271 camera, Tokyo, Japan).

### 4.4. Transcriptome Sequencing

To explore the mechanism that accounted for the difference in proliferative potential of different embryogenic cell lines of Korean pine, we used two EC cell lines F and S as tested materials, performed transcriptomic and metabolomic analyses, and subsequently identified multiple key genes and metabolites regulating EC proliferative potential. After 14 days of culture, EC was collected and quickly frozen in liquid nitrogen and stored in −80 °C refrigerator for transcriptome sequencing analysis. The samples were taken out from the −80 °C freezer and used to extract RNA with the RNA Extraction Kit (DP411, Tiangen, Beijing, China). Then, 1 μL of the RNA solution was used to detect the concentration using Nanodrop. RNA integrity was assessed using the Agilent Bioanalyzer (2100 RNA Nano 6000).

A total of 1 μg of RNA was used for the construction of sequencing libraries with different index barcodes indicating different samples. Finally, the library fragments were purified using the AMPure XP system and the library quality was assessed on the Agilent Bioanalyzer 2100 system. Indexed/barcoded samples were clustered on the cBot cluster generation system using the TruSeq PE Cluster Kit v3-cBot-HS (Illumia). After cluster generation, library preparations were sequenced on the Illumina Hiseq platform and paired-end reads were generated. The experiments were performed in triplicate.

The test data were collated by Excel 2003, and BLAST [43] software was used to align the unigene sequence with Nr [44], Swiss-prot [45], GO [46], COG [47], KOG [48], eggNOG4.5 [49], and Pfam databases. KOBAS 2.0 was used to obtain the KEGG orthology results of unigene. HMMER was used after predicting the amino acid sequence of unigene. The software was compared with the Pfam database to obtain the annotation information of unigene.

### 4.5. qRT-PCR Validation on DEGs

The AAE7 (c154707.graph_c1, Acyl-activating enzyme 7) gene of Korean pine was used as the internal control gene and twelve DEGs were validated by qRT-PCR. The primers were synthesized by Beijing biomarker technologies corporation. The details of the primer sequences are shown in Appendix A (supplementary data at International Journal of Molecular Sciences online). RNA OD value was detected by ultra-trace nucleic acid protein analyzer (scandrop100), and reverse transcription was performed by aidlab company reverse transcription kit (TURE script 1st Stand cDNA SYNTHESIS Kit, Haoxin Biology, Hangzhou, China). PCR instrument (Analytikjena-easycycler, Jena, germany) was used to complete fluorescence analysis and detection. The expression level of the target gene was determined using the 2^−ΔΔCt^ calculation method, and the calculation was performed according to the method of Livak [50], with 3 biological replicates for each treatment.

### 4.6. Metabonomic Analysis

#### 4.6.1. Sample Extraction and Analysis

ECs grown from the F and S cell lines were placed in a lyophilizer (Scientz-100F) for vacuum freeze-drying. After being fully dried, they were thoroughly ground into powder (30 Hz, 1.5 min) with a grinder (MM 400, Retsch, Shanghai, China). A total of 100 mg of powder was dissolved in 1.2 mL of 70% methanol solution for extraction (vortex mixing once every 30 min for 30 s each time, vortexing and mixing 6 times in total). The extracted mixture centrifuged at 12,000 rpm for 10 min, and the supernatant was filtered through a 0.22 μm filter for UPLC-MS/MS analysis (high-performance liquid chromatography: UPLC, SHIMADZU Nexera X2, Tsushima, Japan; ultra and tandem mass spectrometry: MS/MS, Applied Biosystems 4500 QTRAP, Lincolnshire, Nebraska, USA). The experiments were performed in triplicate.

#### 4.6.2. Association Analysis of Metabolomic and Transcriptomic Data

The above transcriptome and metabolite identification results were analyzed jointly. Set the criteria: the screening criteria for differential expression of metabolites with folded difference (FC) > 1, *p* < 0.05, and VIP > 1 and the screening criteria for differential expression of genes with fold difference ≥ 2, FDR ≤ 0.01.

### 4.7. Validation of Important Key Substances Related to the Proliferation of Korean Pine Embryogenic Cells

#### 4.7.1. A Determination of Intracellular Glutathione (GSH) Content in the F and S Cell Lines

The content of GSH was determined by a Nanjing Jiancheng kit (Nanjing Jiancheng Bioengineering Institute, Nanjing, China, GSH A 006-2-1). The model of glutathione oxidized (GSSG) content kit was purchased from Nanjing Jiancheng Bioengineering Institute, A 061-1. The calculation method of total glutathione (T-GSH) content was the sum of GSH and GSSG. GSH: GSSG was described as the ratio of GSH to GSSG.

#### 4.7.2. Determination of Intracellular Hormone Content in the F and S Cell Lines

The contents of indole-3-acetic acid (IAA), zeatin (ZR) and abscisic acid (ABA) in the F and S cell lines were determined by an enzyme-linked immunosorbent assay (ELISA) kit (Shanghai Enzyme-Linked Biotechnology Co., Ltd., Shanghai, China), and the specific assay method was shown in the kit instruction.

#### 4.7.3. Validation of the Effects of Exogenous GSH and BSO on EC Proliferative Rates in the F and S Cell Lines

F and S cell lines EC (0.2 g) were inoculated into the mLV basic medium supplemented with 0.5 mmol·L^−1^ GSH (after cooling in sterile water, put in GSH to dissolve, adjusted pH to 5.8, filtered and sterilized with a filter membrane with a pore size of 0.22 μm, and added to the medium) or 0.5 mmol·L^−1^ BSO (sterilized the medium and cooled it to about 55–60 °C, and filtered it through a filter membrane with a pore size of 0.22 μm), the treatment without GSH and BSO was used as the control (CK), the fresh weight was measured on the 14th day of EC proliferation culture, and the EC proliferative rate was calculated.

### 4.8. Data Collection and Analysis

Experimental calculations, including proliferation parameters, were performed using Excel 2003 (Microsoft, Redmond, DC, USA). A one-way analysis of variance (ANOVA) and Duncan’s multiple comparisons tests were performed using SPSS 19 (IBM, Armonk, NY, USA), mean ± se, *n* = 5. Graphs were constructed in Sigma Plot 12.0 (Systat, Chicago, IL, USA).

## 5. Conclusions

There were differences in proliferative potential among different embryogenic cell lines of Korean pine. Integrated transcriptome–metabolome analysis shows that glutathione-related genes and metabolites play a positive role in the proliferation of embryogenic callus of Korean pine. Through the analysis results, it was found that the glutathione content in the F (Fast proliferative rate) cell line was significantly higher than that in the S (Slow proliferative rate) cell line. When exogenous glutathione was added to the two cell lines, the proliferation rate of both cell lines was promoted, which further confirmed the results of transcriptome and metabolome. Our research laid a foundation for revealing the regulatory mechanism of woody plant cell proliferation and cell totipotency expression, established the technology to improve woody plant embryogenic callus proliferative ability, optimized the woody plant somatic embryogenesis system, and provided a new ground for revealing the molecular regulatory mechanism of conifer somatic embryogenesis.

## Figures and Tables

**Figure 1 ijms-23-14679-f001:**
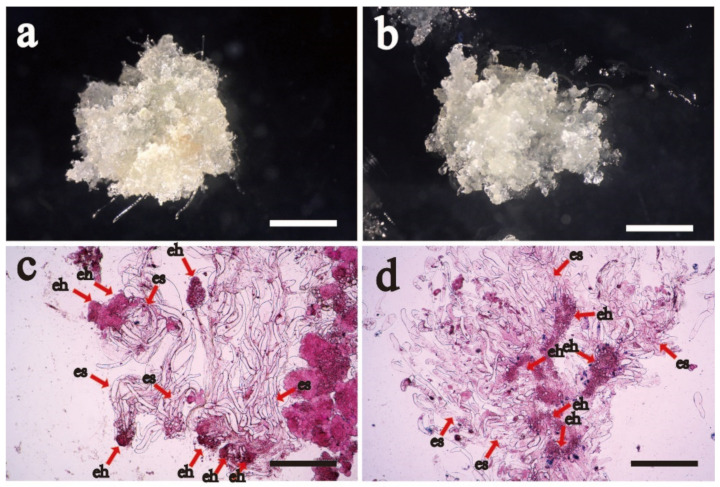
Macrostructural and cytological observation of embryogenic callus in F and S Korean pine cell lines. Note: (**a**) F cell line macrostructure, bar = 1 cm; (**b**) S cell line macrostructure, bar = 1 cm; (**c**) F cell line cytological structure, bar = 150 μm; and (**d**) S cell line cytological structure, bar = 150 μm. The letters eh denote the embryo head (eh), and the letter s denotes the suspensor (es).

**Figure 2 ijms-23-14679-f002:**
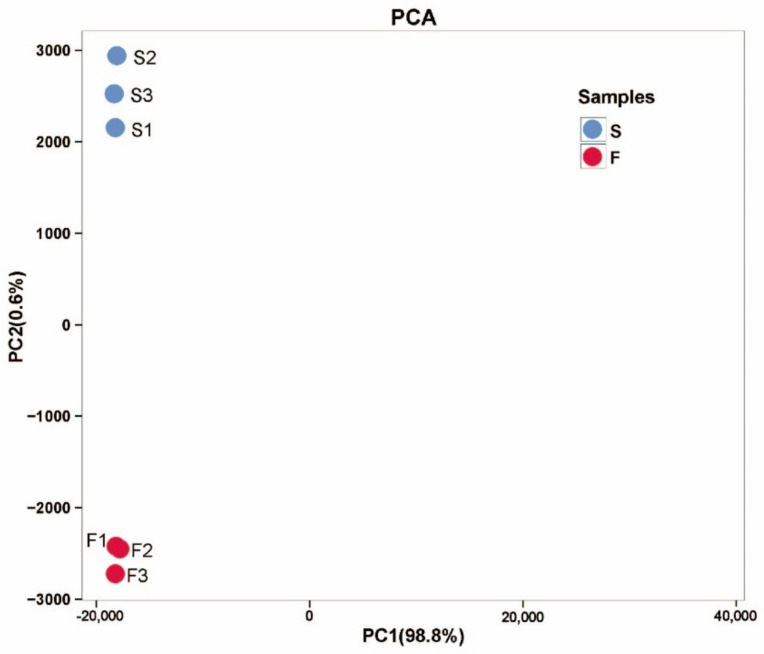
PCA analysis of the principal components in F and S cell lines of Korean pine. Note: The X-axis represents the first principal component, and the Y-axis represents the second principal component.

**Figure 3 ijms-23-14679-f003:**
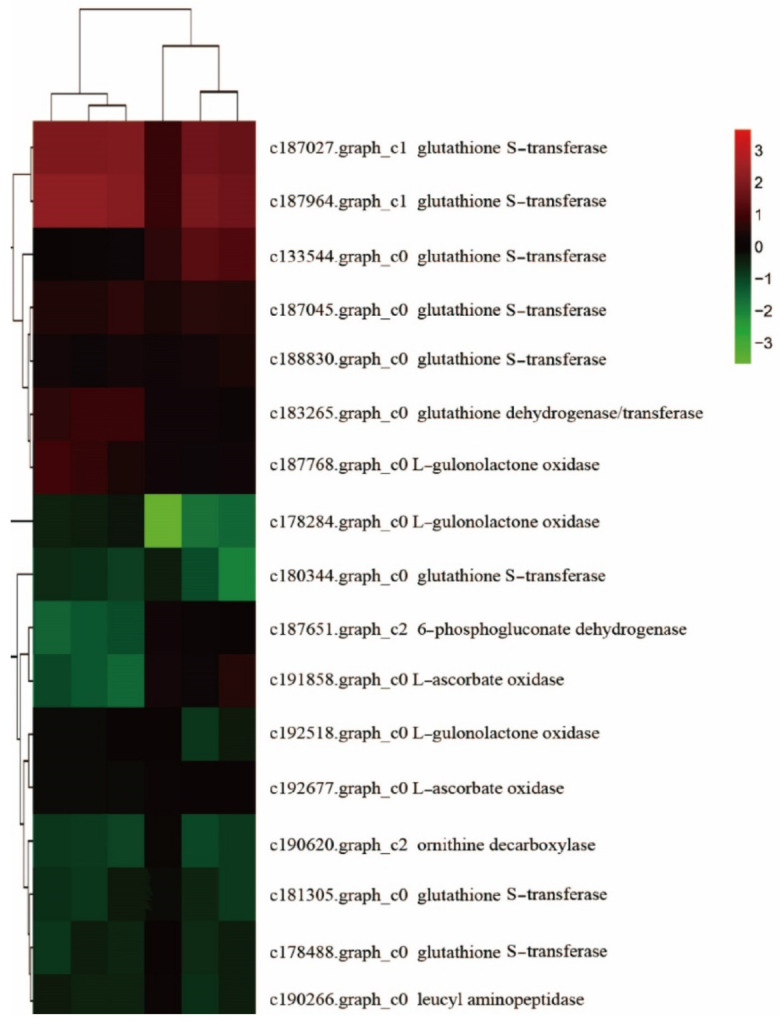
Cluster heat map of GSH-related differential genes between F and S cell lines of Korean pine. Note: F1, F2, and F3 are three biological replicates of the F cell line; S1, S2, and S3 are three biological replicates of the S cell line, The note of Figure 4 and Figure 8 are the same as Figure 3.

**Figure 9 ijms-23-14679-f009:**
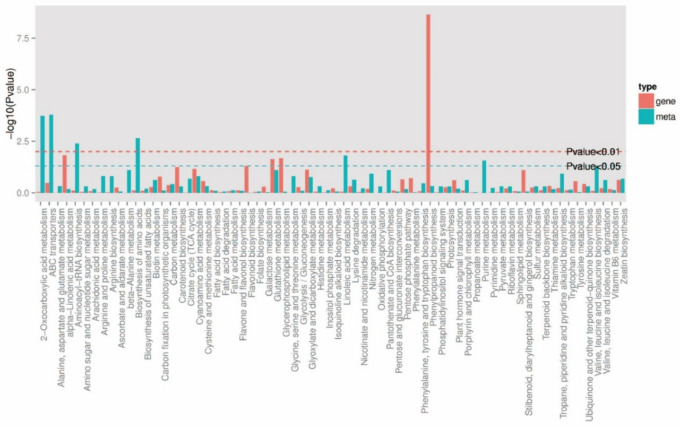
DEMs and DEGs co-enrichment map in F and S cell lines of Korean pine.

**Figure 10 ijms-23-14679-f010:**
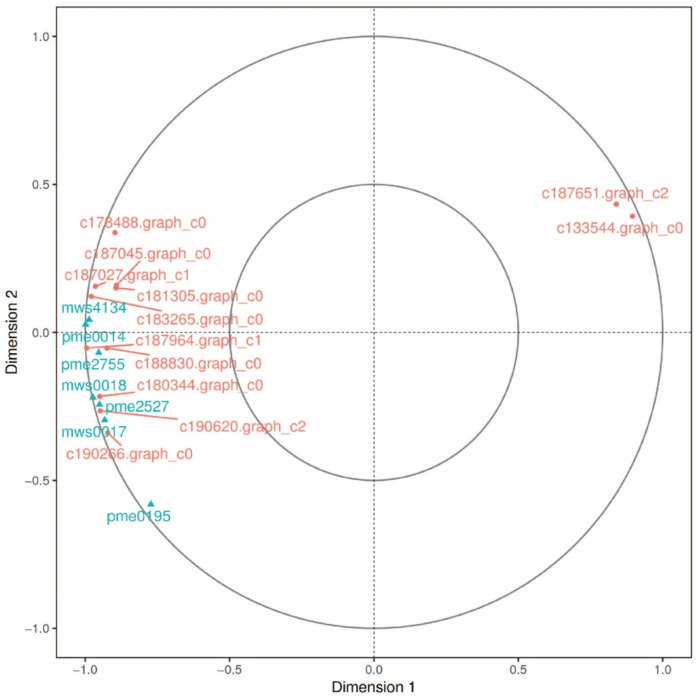
Canonical correlation analysis of Korean pine F and S cell lines and related gene metabolites of GSH. Note: The figure is divided into four regions. The farther away from the origin, the higher correlation will be in the same region, and there is a high degree of correlation between genes and metabolites close to each other in the same region.

**Figure 11 ijms-23-14679-f011:**
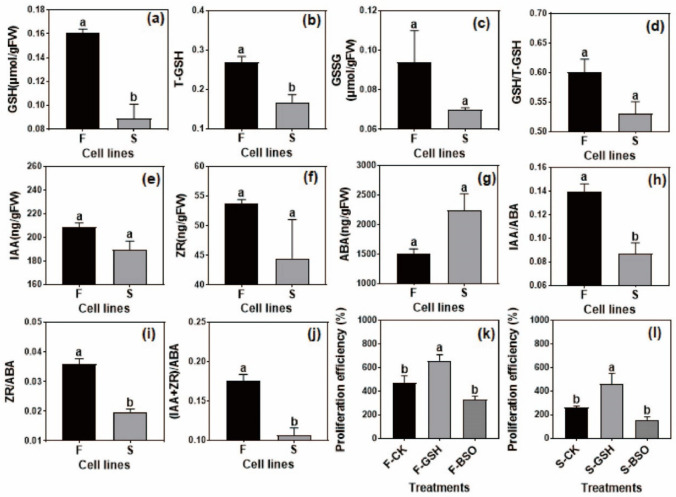
Intracellular GSH, hormones content, and fresh weight proliferation in F and S cell lines of Korean pine. Note: The average 3–4 biological replicates were counted for each treatment; ANOVA and Duncan’s tests were performed on the data in the figure, mean ± se, *n* = 5. Different lowercase letters at the same little figure indicate significant differences (*p* ˂ 0.05), (**a**–**d**) shows the content of glutathione in F and S cell lines, (**e**–**j**) shows the content of plant growth regulators in F and S cell lines, (**k**,**l**) shows the proliferation capacity of F and S cell lines exogenous addition of GSH and BSO.

**Figure 12 ijms-23-14679-f012:**
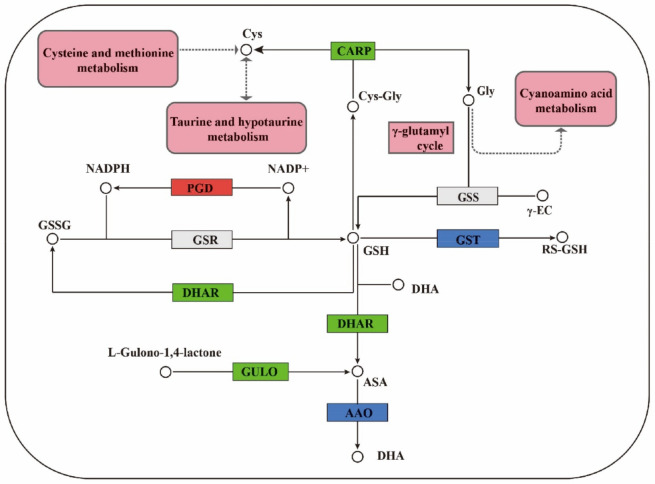
Diagram of the GSH cycle pathway in F and S cell lines of Korean pine. Note: Cys: L-cysteine; Cys-gly: L-cysteinylglycine; Gly: glycine; RS-GSH: R-S-glutathione; γ-GT: gamma-glutamyltranspeptidase; γ-EC: γ-glutamylcysteine; RS-GSH: R-S-glutathione; GSH: Glutathione; GSSG: oxidized glutathione; ASA: ascorbic acid; DHA: dehydroascorbic acid.

**Figure 13 ijms-23-14679-f013:**
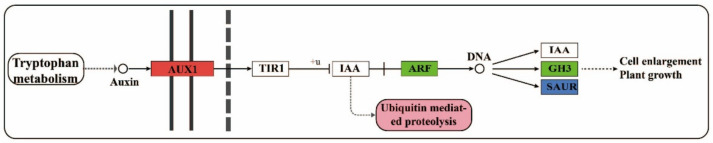
Diagram of auxin signal transduction pathway in F and S cell lines of Korean pine.

## Data Availability

The data presented in this study are available in this manuscript.

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
