# Peer review of "Glutathione Plays a Positive Role in the Proliferation of Pinus koraiensis Embryogenic Cells"

_ijms, 2022, doi:10.3390/ijms232314679_

Round 1

Reviewer 1 Report

I think this article should be rewritten. The authors performed many analyses and obtained a big amount of experimental data. However the analysis and discussion of these data are quite poor. At the moment the article contains many interesting facts but these facts are often not sufficiently connected with each other in Discussion and conclusion. Maybe it would be better to divide the article to two ones and that allows analyzing the collected data much more deeply.

Author Response

Response to Reviewer 1 Comments

Point 1: I think this article should be rewritten. The authors performed many analyses and obtained a big amount of experimental data. However the analysis and discussion of these data are quite poor. At the moment the article contains many interesting facts but these facts are often not sufficiently connected with each other in Discussion and conclusion. Maybe it would be better to divide the article to two ones and that allows analyzing the collected data much more deeply.

Response 1:

Dear reviewer, thank you very much  for your valuable comments on the manuscript. We have revised and improved the manuscript comprehensively, including materials and methods, results and analysis, rewritten the discussion part, simplified the conclusion part, and revised and improved the format of the references comprehensively, all of which have been marked in the paper.

This manuscript is not suitable to be divided into two articles, because if it is divided into two articles, one article only include transcriptomics, metabolomics and qRT-PCR verification, while the other article only include a small part of physiological data and embryonic callus proliferation ability observation data. In fact, all the experiments in the manuscript are actually inseparable, and we deduce from the molecular biology data that GSH may be the key substance affecting the proliferation of red pine embryonic calluses, and we have further conducted more in-depth research, including the response of embryonic callus proliferation after exogenous addition of GSH and GSH synthesis inhibitor BSO, as well as the determination of intracellular GSH, GSSG, T-GSH and plant growth regulatory substances, confirming the results of molecular biology speculation. That is Glutathione plays a positive role in the proliferation of Pinus koraiensis embryogenic cells, so we believes that the contents of the manuscript are interlinked and inseparable.

Reviewer 2 Report

Recommendations and comments on the review ijms-1983656-peer-review: Glutathione plays a positive role in the proliferation of Pinus koraiensis embryogenic cells” 

Pine is an important tree species that plays a critical role in the wood demand of the global wood industry and the country’s economy.   Biotechnology tools can speed up the rapid production of a large number of regenerated plants pine through asexual propagation. In the presented research, the authors well planned this study to understand the role of Glutathione (GSH) in the proliferation of Pinus koraiensis embryogenic cells. Its results clearly showed upregulated expression of GSH in the F cells line. The results of this study will be helpful to further investigate the role of GSH in the proliferation of embryogenic cells in pine and other wood crops. The Manuscript is well scientifically conducted and covered in the M&M, research, and discussion sections and reported. The Manuscript requires a minor revision to accept for publication.

1. Introduction

Line 30: Make a space between growincrease; grow increase

Line 66 to 67: Authors need to explain the meaning of F cells and S cells in text for reader’s clarity? Or authors use these words to indicate two different embryogenic cells.

2. Results

Why some heading/titles of Result’s section are in italics (Line 76, 92) and mostly headings are in non-italics? Authors need to use same pattern across the manuscript. 

Provide a space between heading number and its title text from 2.2.5.1 to 2.2.5.4. There is no space between heading number and its text. For example: 2.2.5.1KEGG enrichment analysis in differentially expressed genes. The correct one is 2.2.5.1 KEGG enrichment analysis in differentially expressed genes 

In Figure 3, 4 and 9 of cluster heat map, what is F1, F2, F3 and S1, S2, S3? Are these replication or different embryogenic cells. Authors need to add more text in title of these figures for better understanding of readers. 

In Figure 11, Authors need to add more text in its heading to explain this figure for easy understanding. For example, what is meaning of a and b on top of F and S cell bar?

4. Materials and methods

In this section, mostly headings are italics, but few are non-italics. Authors need to use same pattern across the manuscript. 

Author Response

Dear Reviewer,

Our sincere thanks to you for the time and effort that you have put into reviewing our manuscript! We found all the comments very constructive and helpful, and have revised our manuscript according to all comments. Please find, below, our point-by-point response to the comments raised.

Thank you for considering our revised manuscript!

Point 1: Line 30: Make a space between growincrease; grow increase

Response 1: We have separated “grow” and “increase” .

Point 2: Line 66 to 67: Authors need to explain the meaning of F cells and S cells in text for reader’s clarity? Or authors use these words to indicate two different embryogenic cells.

Response 2: The meaning of F and S cell lines has been added to the text:Fast proliferative potential cell line is abbreviated as F; Slow proliferative potential cell line is abbreviated as S.

Point 3: Why some heading/titles of Result’s section are in italics (Line 76, 92) and mostly headings are in non-italics? Authors need to use same pattern across the manuscript.

Response 3: According to the formatting requirements of Int. J. Mol. Sci., the secondary headings are italics and the tertiary headings are non-italics, and the full text headings have been revised and unified.

Point 4: Provide a space between heading number and its title text from 2.2.5.1 to 2.2.5.4. There is no space between heading number and its text. For example: 2.2.5.1KEGG enrichment analysis in differentially expressed genes. The correct one is 2.2.5.1 KEGG enrichment analysis in differentially expressed genes 

Response 4: A space has been added between the title and the number.

Point 5: In Figure 3, 4 and 9 of cluster heat map, what is F1, F2, F3 and S1, S2, S3? Are these replication or different embryogenic cells. Authors need to add more text in title of these figures for better understanding of readers. 

Response 5: Figure 3,4 and 9 have been added to the figure notes: Note: F1, F2, F3 are three biological replicates of F cell line; S1, S2, S3 are three biological replicates of S cell line, Figure 4 and 8 are the same as Figure 3.

Point 6: In Figure 11, Authors need to add more text in its heading to explain this figure for easy understanding. For example, what is meaning of a and b on top of F and S cell bar?

Response 6: Figure 5 and 11 have been added to the figure notes.

Figure 5, Note: The average 3 biological replicates was counted for each treatment, ANOVA were performed on the data in the figure, mean ± se.

Figure 11, Note: The average 3-4 biological replicates was counted for each treatment, ANOVA and Duncan's test were performed on the data in the figure, mean ± se, N= 5. Different lowercase letters at the same little figure indicate significant differences (p Ë‚ 0.05).

Point 7: 4. Materials and methods: In this section, mostly headings are italics, but few are non-italics. Authors need to use same pattern across the manuscript. 

Response 7: According to the formatting requirements of Int. J. Mol. Sci., italics is used for secondary headings and non-italics is used for tertiary headings, and the title of the full text has been revised to be uniform.

Reviewer 3 Report

Lines 65 – 69:……To explore the mechanism that acco……………proliferative potential: Move to Materials and Methods and insert the objective of the study in a very clear and concise words/sentence.

Lines 69 – 74: …The results of transcriptome…….. – move to results and discussion sections.

Line 29: growincrease – recast ‘increase’

Mat & methods

Line 477: Do not start a paragraph with figure 0.42 g. Recast.

Line 485: Change 4.8 to Data collection and analysis. Also expand the section to describe fully the procedures followed in analyzing the data.

Line 369 Discussion: ……….This study found that the expression of Aux/IAA gene trended towards down regulation in S cell line compared with F cell line……….Discuss effectively the gene expression of Aux/IAA and its effect on EC induction and proliferation in P. koraiensis and other related species.

Author Response

Dear Reviewer,

Our sincere thanks to you for the time and effort that you have put into reviewing our manuscript! We found all the comments very constructive and helpful, and have revised our manuscript according to all comments. Please find, below, our point-by-point response to the comments raised.

Thank you for considering our revised manuscript!

Point 1: Lines 65-69:……To explore the mechanism that acco……………proliferative potential: Move to Materials and Methods and insert the objective of the study in a very clear and concise words/sentence.

Response 1: We have moved ......To explore the mechanism that acco............... proliferative potential to the Materials and Methods section.

Point 2: Lines 69 – 74: …The results of transcriptome…….. – move to results and discussion sections.

Response 2: We have moved …The results of transcriptome…….. to results and discussion sections.

Point 3: Line 29: growincrease – recast ‘increase’

Response 3: We have separated “grow” and “increase” .

Point 4: Mat & methodsPoint 1: Line 477: Do not start a paragraph with figure 0. 2 g. Recast.

Response 4: We have transferred 0.2g to another location.

Point 5: Line 485: Change 4.8 to Data collection and analysis. Also expand the section to describe fully the procedures followed in analyzing the data.

Response 5: 4.8 has been changed to Data collection and analysis, and the description has been refined.

Point 6: Line 369 Discussion: ……….This study found that the expression of Aux/IAA gene trended towards down regulation in S cell line compared with F cell line……….Discuss effectively the gene expression of Aux/IAA and its effect on EC induction and proliferation in P. koraiensis and other related species. 

Response 6: Aux/IAA gene has been added to the discussion of the effect on cell proliferation (The plant hormone auxin modulates cell proliferation and cell expansion in part by changing gene expression. Primary auxin response genes consist of members of three gene families: Aux/IAA, GH3 and SAUR [40]. The IAA regulates many aspects of plant growth and development. These processes are controlled by auxin-mediated changes in cell division, expansion, and differentiation [41]. This study found that the expression of Aux/IAA gene trended towards downregulation in S cell line compared with F cell line, Aux/IAA gene promote cell expansion [41], therefore, this may be one of the reasons for the rapid proliferation of F cell lines.)

Reviewer 4 Report

The authors present a sound and important contribution to pine propagation that should be made availabe to the community. There are, however, several suggestions that should be taken into account prior to publication. These are meant to improve readability and, thus, impact of this valuable contribution. 

* Abstracts and similar paragraphs must be readable and understandable as stand-alone versions. The reviewer recommends to avoid abbreviations (here DEG; GSH) and to check for correct writing (here: Buthathionine must read butathionine).

The reviewer recommends, also to check the complete text for unnecessary abbreviations. In any case, abbreviated terms must be mentioned with full name when mentioned first in a paragraph. This very easy measure renders  publications easier to read. It would also be helpful, very shortly to mention the meaning of F and S cell lines. Abstracts are not exclusively meant for experts, but should draw attention of a broader readership.

* Keywords. Please provide keywords with a logical order, alphabethic order would be sufficient. 

* Introduction. Please check for typos and similar throughout the text. Here, as an example: "continues to growincrease". "Sembryogenesis"? There is more. 

* Results. Please check measurements and data for reasonable number of digits. Example: "740.00 ± 51.96" is certainly not reasonable. Also the last digit given must mean something. There is more. 

There are also typos in this paragraph. Examples (there is more): "De novo" should read 'de novo'; why do you write "Clean Data" instead of the usual 'clean data'?

* The authors are asked to check figure captions for clarity. Captions should always be understandable without reference to the main text. Check for unnecessary abbreviations and provide the main result of the data presented.

* Methods. Please check again, if procedures and techniques that have not been developed by the authors themselves are given together with the original citation.

* Conclusion. Present tense must be used for describing your present results, actually throughout the manuscript (please check). Follow the same advices for readability as for the Abstract. 

* References. Please provide every detail exactly as demanded by Journal instructions. 

The reviewer will be glad to promote the contribution for publication after these more merely formal suggestions have been considered. 

Author Response

Dear reviewer,

Thank you very much for your valuable comments on the manuscript. We have revised and improved the manuscript comprehensively.

Point 1: Abstracts and similar paragraphs must be readable and understandable as stand-alone versions. The reviewer recommends to avoid abbreviations (here DEG; GSH) and to check for correct writing (here: Buthathionine must read butathionine).

Response 1: The abbreviations in the abstract have been deleted according to the reviewer 's opinion. In addition, “Buthathionine”was changed to“butathionine”and the modified part was marked with purple.

Point 2: The reviewer recommends, also to check the complete text for unnecessary abbreviations. In any case, abbreviated terms must be mentioned with full name when mentioned first in a paragraph. This very easy measure renders publications easier to read. It would also be helpful, very shortly to mention the meaning of F and S cell lines. Abstracts are not exclusively meant for experts, but should draw attention of a broader readership.

Response 2: According to the reviewer 's recommendation, unnecessary abbreviations have been removed from the manuscript and full names are noted when abbreviations first appear in this manuscript.

Point 3: Keywords. Please provide keywords with a logical order, alphabethic order would be sufficient. 

Response 3: According to the reviewer 's recommendation, we put the keywords in alphabetical order.

Point 4: Introduction. Please check for typos and similar throughout the text. Here, as an example: "continues to growincrease". "Sembryogenesis"? There is more. 

Response 4: The introduction has been checked and "growincrease" has been changed to "grow increase". "Sembryogenesis" has been changed to "Somatic embryogenesis". "xanthine nucleosides and methyloxazole" has been changed to "xanthosine and methyloxazole".

Point 5: Results. Please check measurements and data for reasonable number of digits. Example: "740.00 ± 51.96" is certainly not reasonable. Also the last digit given must mean something. There is more. 

Response 5: According to the reviewer 's recommendation, The results and analysis section of the decimal place to retain a comprehensive modification, and the modified part was marked with purple.

Point 6: There are also typos in this paragraph. Examples (there is more): "De novo" should read 'de novo'; why do you write "Clean Data" instead of the usual 'clean data'?

Response 6: The results and analysis have been comprehensively checked according to the reviewer 's recommendation, such as modifying "De novo" to "de novo", "Unigene" to "unigene", etc., and marking with purple fonts.

Point 7: The authors are asked to check figure captions for clarity. Captions should always be understandable without reference to the main text. Check for unnecessary abbreviations and provide the main result of the data presented.

Response 7: According to the reviewer 's recommendation, we have checked all the drawings in this manuscript, and all drawings were greater than 300dpi. We have also checked all the drawings and modified them. In addition, all abbreviations in the text have been checked, unnecessary abbreviations have been removed and marked them with purple.

Point 8: Methods. Please check again, if procedures and techniques that have not been developed by the authors themselves are given together with the original citation.

Response 8: References have been cited in the materials and methods based on reviewer comments.

Point 9: Conclusion. Present tense must be used for describing your present results, actually throughout the manuscript (please check). Follow the same advices for readability as for the Abstract. 

Response 9: The acronyms in the conclusion have been removed and the conclusion have been simplified according to the reviewer's opinion.

Point 10:* References. Please provide every detail exactly as demanded by Journal instructions. 

Response 10: The references have been comprehensively modified and improved according to the reviewer's opinion.

Round 2

Reviewer 1 Report

Discussion has been improved, although it would be better if it were deeper